# Health extension workers' perceived health system context and health post preparedness to provide services: a cross-sectional study in four Ethiopian regions

Theodros Getachew [1,2] Solomon Mekonnen Abebe [2] Mezgebu Yitayal,[2] Anna Bergström,[3] Lars-Ake Persson [1,4] Della Berhanu [1,4]

► Prepublication history and supplemental material for this paper is available online. To view these files, please visit the journal online (http://dx.doi.org/10.1136/bmjopen-2020-048517).

[1]Health System and Reproductive Health Research Directtorate, Ethiopian Public Health Institute, Addis Ababa, Ethiopia
[2]Institute of Public Health, University of Gondar College of Medicine and Health Sciences, Gondar, Ethiopia
[3]Uppsala University, Uppsala, Sweden
[4]London School of Hygiene and Tropical Medicine, London, UK

**Correspondence to**
Theodros Getachew;
tedi.getachew@yahoo.com

## ABSTRACT

**Objective** The health system context influences the implementation of evidence-based practices and quality of healthcare services. Ethiopia aims at reaching universal health coverage but faces low primary care utilisation and substandard quality of care. We assessed the health extension workers' perceived context and the preparedness of health posts to provide services.

**Setting** This study was part of evaluating a complex intervention in 52 districts of four regions of Ethiopia. This paper used the endline data collected from December 2018 to February 2019.

**Participants** A total of 152 health posts and health extension workers serving selected enumeration areas were included.

**Outcome measures** We used the Context Assessment for Community Health (COACH) tool and the Service Availability and Readiness Assessment tool.

**Results** Internal reliability of COACH was satisfactory. The dimensions *community engagement*, *work culture*, *commitment to work* and *leadership* all scored high (mean 3.75–4.01 on a 1–5 scale), while *organisational resources, sources of knowledge* and *informal payments* scored low (1.78–2.71). The general service readiness index was 59%. On average, 67% of the health posts had basic amenities to provide services, 81% had basic equipment, 42% had standard precautions for infection prevention, 47% had test capacity for malaria and 58% had essential medicines.

**Conclusion** The health extension workers had a good relationship with the local community, used data for planning, were highly committed to their work with positive perceptions of their work culture, a relatively positive attitude regarding their leaders, and reported no corruption or informal payments. In contrast, they had insufficient sources of information and a severe lack of resources. The health post preparedness confirmed the low level of resources and preparedness for services. These findings suggest a significant potential contribution by health extension workers to Ethiopia's primary healthcare, provided that they receive improved support, including new information and essential resources.

## Strengths and limitations of this study

► This study was the first assessment of Ethiopian frontline primary healthcare workers' perceived health system context and the facility preparedness for services.
► Understanding the health system context increases the likelihood of successful implementation of evidence-based practices.
► The Context Assessment for Community Health tool has been validated in a range of other low-income countries and was found to have satisfactory internal reliability when translated into three Ethiopian languages.
► Although precautions were taken to obtain valid responses from the interviewed health extension workers, we cannot exclude the risk of respondents' bias.
► The sample represented 52 districts in four Ethiopian regions that participated in a child health services study, and inferences cannot be drawn to the whole country.

## INTRODUCTION

The health system context is essential for new interventions and quality of care.[1] Healthcare of poor quality contributes to the high mortality in low-income and middle-income countries.[2 3] These quality problems have multiple causes, for example, lack of resources and suboptimal interaction between healthcare providers and clients.[4] Thus, there is a growing understanding that the health system context matters for efforts to improve health services quality.[5 6] However, we lack consensus on the definition, operationalisation and methods to study context.[1]

Therefore, we need systematic ways of assessing the context in which health workers are active.[7] The most frequently used tools and information systems provide structural

information, for example, the Service Availability and Readiness Assessment tool.[8] Mapping the facility preparedness sets the scene, but such assessments are poorly associated with the quality of services provided.[9] A meta-review showed that access to information, community engagement, leadership, regulations and standards, organisational capacity, models of care, communication, and work satisfaction are essential for implementing new interventions and quality of care.[6 10] The Context Assessment for Community Health (COACH) tool was developed and validated in low-income and middle-income countries and included many of the context dimensions mentioned above.[11]

In Ethiopia, primary-level services for under-5 children are provided through the health extension programme.[12 13] This programme is a community-based strategy to expand access to basic health promotion, disease prevention and selected curative health services.[14] The programme is operated by the health extension workers at the community level.[15] Two female community health workers, known as health extension workers, provide preventative and curative services for a population of 5000. They offer static services from health posts as well as outreach services within the community.[16] The health extension workers are recruited from the community they serve and deployed to service after a 1-year formal preservice training provided after completing 10th grade of formal education.[14] Five health posts, their referral health centre and a primary district-level hospital comprise the primary healthcare unit.[17] Health posts are the most peripheral units, providing mainly preventive care and selected curative services.[18] Despite the successful implementation of the health extension programme, the programme is currently facing challenges that remain to be addressed. These challenges are related to the utilisation and quality of services offered by the health extension workers and their working and living conditions.[19]

The Ethiopian Ministry of Health aims to increase the primary healthcare services' access and quality through reforms and new initiatives.[20 21] One such effort was the Optimizing the Health Extension Program intervention to increase the quality and utilisation of health services for under-5 children. As part of the evaluation of that intervention, we have shown that health extension workers did not follow the clinical guidelines for assessing and managing sick children with common illnesses.[22] Their ability to classify childhood illnesses was also low.[23] Unfortunately, the intervention, which included community engagement, training, supportive supervision and performance reviews of health workers neither increased care-seeking for sick children[24] nor improved the classification of childhood illnesses by these primary healthcare workers.[25] The failure of such an intervention could, at least partly, be attributed to the context of the health extension workers. There is a need for accurate measurements that reflect the health system context in which care is provided to patients and populations.[26] Therefore, we aimed to assess the health extension workers' perceived health system context and the health posts' service readiness in four Ethiopian regions.

## METHODS
### Study setting and design
This study was part of a large project, which assessed a complex intervention's effectiveness to increase care-seeking for children under 5 years. This intervention had three components: community engagement, capacity building, and ownership and accountability of child health services. The assessment was done in 52 districts of four regions (Amhara; Tigray; Oromia; Southern Nations, Nationalities, and Peoples) of Ethiopia with baseline and endline surveys conducted before and after

**Table 1** Definitions of context assessment dimensions in the Context Assessment for Community Health tool

| Dimensions | Number of items | Definition |
| --- | --- | --- |
| Resources | 11 | The availability of resources (staff, space, time, communication and transport, drugs, equipment and supplies, finance) that allows a unit to adapt successfully to internal and external pressures. |
| Community engagement | 5 | The mutual communication, deliberation and activities that occur between community members and units. |
| Monitoring services for action | 5 | The process of using data to assess group/team performance. |
| Sources of knowledge | 5 | The structures that facilitate the ability to access and use knowledge. |
| Commitment to work | 3 | The relative strength of an individual's identification with and involvement in a particular work organisation. |
| Work culture | 6 | The way that 'we do things' in our organisations and work units. This includes culture of learning and change, and culture of responsibility. |
| Leadership | 6 | The actions of formal leaders in an organisation (unit). |
| Informal payment | 8 | Payments to individuals, which are made outside official payment channels including nepotism and accountability. |

**Table 2** Domains and their items used to construct the health post service availability and readiness

| Domain | Items |
|---|---|
| Basic amenities | Communication equipment |
| | Access to adequate sanitation facilities for clients |
| | Improved water source |
| | Power supply |
| Basic equipment | Infant scale |
| | Thermometer |
| | Functional stethoscope |
| | Mid-upper arm circumference tape measure |
| Standard precautions | Sharps container |
| | Chlorine bleach |
| | Bucket for decontamination solution |
| | Contaminated waste container |
| | Soap and towel or hand rub |
| | Alcohol-based hand rub |
| | Clean gloves |
| Diagnostics | Malaria rapid diagnostic test |
| Essential medicines | Vitamin A |
| | Gentamycin |
| | Amoxicillin susp/tab |
| | Oral rehydration solution |
| | Zinc |
| | Coartem (artemether/lumefantrine) |
| | Ready-to-use therapeutic food |

the intervention. This paper used the endline data that were conducted from December 2018 to February 2019. The protocol and results of the evaluation have been published.[24 27]

### Subjects

A total of 200 enumeration areas were selected to represent the selected districts in the endline survey. Health posts serving these areas were included in the study, and their preparedness for services was assessed. One health extension worker at each health post was interviewed, and their perceived context was evaluated. We considered datasets with information from health posts as well as their respective health extension workers.

### Study tools

Two tools were used at facility and provider levels. The provider-level tool aimed to assess the health extension workers' perceived context on the service delivery environment. The tool, labelled the COACH, has 49 items that measure eight dimensions of context (table 1).[28] The tool was developed in Bangladesh, Vietnam, Uganda, South Africa and Nicaragua.[11] It also includes

demographic questions on age, gender, professional qualification, health facility and years working at the current facility. The tool items were measured on a 5-point Likert scale ranging from 'strongly disagree' to 'strongly agree'. Items in *source of information* were measured for availability and frequency of use. The Brislin model[29] was used to translate the tool into Amharic, Oromiffaa and Tigrigna, including forward translation, review of the translated tool, backward translation, and comparison of the original and back-translated tools. The forward translation was done by a professional translator. The review, backward translation and comparisons were done by a group of experts, including the study team. Conrad and Blair's taxonomy[30] was used to describe the problems that appeared in the translations. Accordingly, there were six lexical problems with difficulties in the meanings of words, one logical problem, and one inclusion or exclusion problem. All identified translation problems were possible to correct.

The facility tool aimed at collecting information on the overall facility-level preparedness to provide child health services (table 2), which was based on the WHO Service Availability and Readiness Assessment reference manual.[31] The tool was translated into three local languages (Amharic, Oromiffaa and Tigrigna), pretested and amended.

### Measurements

We judged the COACH tool's internal consistency with Cronbach's alpha[32] that expressed if items in the instrument's different dimensions measured the same thing. Descriptive statistics were used to assess the health extension workers' agreement to the items and dimensions. All items except for source of knowledge were measured on a scale of 1–5, where the scores for items 48 and 49 were reversed to measure in the same direction as other items. The overall agreement was a calculated value drawn by multiplying the number of items in the dimension by four, which was coded as agreement. An individual was considered to agree if her score was above the calculated value.

The general health service readiness score was a composite summary measure calculated by combining information from the five general service readiness domains: basic amenities, standard precautions for infection prevention, basic equipment, diagnostics and essential medicines.[31] For each domain, the average availability of tracer items was revealed as the domain score. Each dimension's mean score was computed to assess the average responses to the included items in the dimension. The analysis was performed using STATA V.14.2 statistical package (Stata Corp, College Station, Texas, USA).

### Patient and public involvement

Patients or the public were not involved in the design or conduct, or reporting or dissemination plans of this research.



**Table 3** Summary of perceived context of health extension workers and the internal consistency of the Context Assessment for Community Health tool; survey in four Ethiopian regions, 2018 (N=152)

| Dimensions | Number of items | Mean (SD) | Cronbach's alpha | Average interitem correlation |
|---|---|---|---|---|
| Resources | 11 | 2.60 (0.60) | 0.7620 | 0.2255 |
| Community engagement | 5 | 4.01 (0.58) | 0.8813 | 0.5975 |
| Monitoring services for action | 5 | 3.75 (0.70) | 0.8678 | 0.5676 |
| Sources of knowledge | 5 | 2.71 (0.79) | 0.5053 | 0.1696 |
| Commitment to work | 3 | 3.79 (0.79) | 0.7976 | 0.5677 |
| Work culture | 6 | 3.89 (0.51) | 0.7683 | 0.3559 |
| Leadership | 6 | 3.79 (0.60) | 0.8771 | 0.5432 |
| Informal payment | 8 | 1.78 (0.56) | 0.8427 | 0.4011 |

## RESULTS

Of the 200 enumeration areas, 20 were not included due to local unrest. The remaining 180 enumeration areas were served by 165 health posts. A total of 165 health posts were assessed, and 154 health extension workers were available for interview. Eleven health posts did not have data on their respective health extension workers and two health extension workers were interviewed without their respective health post data. After merging the two datasets, 152 health post and health extension worker data were available for analysis.

### Perceived context

Table 3 presents the average interitem correlation and the Cronbach's alpha coefficients for the eight context dimensions. Almost all dimensions exceeded the commonly accepted standard for satisfactory internal reliability (0.70) for new scales (α range=0.51–0.89). One dimension (*source of knowledge*, α=0.51) did not meet this standard. The average interitem correlation ranged from 0.17 to 0.59. The ideal range of average interitem correlation is 0.15–0.50; less than 0.15 indicates that items are not well correlated and do not measure the same idea very well. More than 0.50 means that items are close, almost repetitive.

The mean scores of the COACH dimensions on a scale of 1–5 are presented in table 3. The dimensions *community engagement, work culture, commitment to work* and *leadership* all scored high (mean 3.75–4.01 on the 1–5 scale), while *organisational resources, sources of knowledge* and *informal payments* scored low (1.78–2.71). These findings indicate that the health extension workers neither perceived themselves as having sufficient resources to conduct their work nor to have access to new knowledge.

Tables 4 and 5 depict the percentage of each item included in the eight context dimensions. Most of the health extension workers reported disagreement on the availability of financial resources. They also disagreed to having access to communication and transport.

Figure 1 depicts the percentage of average scores for the context dimensions. Very few (2.6%) perceived their facility to have enough resources available to manage their work. Most respondents (83.6%) perceived that their facility had active communication with members of their communities. Sixty-six per cent on average responded agreement for the *work culture* dimension, implying that they considered their *work culture* to support learning, change and responsibility. A very high proportion of respondents (98.7%) regarded *informal payment* for health workers not to be acceptable in their facility.

There was no difference in context dimensions between intervention and comparison areas in the evaluation's endline survey (all p>0.05). The exact percentage for each item is found in online supplemental table S1.

### General facility-level readiness

Figure 2 shows the general service readiness index and domain scores. The general service readiness index was 59%, implying that 6 in 10 health posts were ready to provide child health services. On average, about two-thirds (67%) of health posts had basic amenities to provide services, 81% had basic equipment required, 42% had standard precautions for infection prevention, 47% had diagnostic test capacity for malaria rapid diagnostic test and 58% had essential medicines. The basic equipment mean score was the highest across the five domains, and the diagnostic mean score index was the lowest.

Figure 2 also shows the percentage of health posts having all tracer items available to provide general child health services. Accordingly, only 1% of health posts had all essential medicines. Half of the health posts had all tracer items for basic equipment. Three in 10 health posts had all items for basic amenities.

## DISCUSSION

We have described the Ethiopian health extension workers' perceived context and the health posts' preparedness to provide child health services. The health extension workers perceived that they had a good relationship with the local community. They were active in using data for planning and performance, were highly committed to their work and had positive perceptions of their work culture. They also had a relatively positive experience of their leaders and reported no corruption or informal payments. In contrast, they reported having insufficient

**Table 4** Percentage of items and dimensions of the Context Assessment for Community Health tool in four Ethiopian regions, 2018 (N=152)

| Resource | Disagree | Neutral | Agree |
|---|---|---|---|
| 1. My unit has enough workers with the right training and skills to do everything that needs to be done. | 52 | 3 | 45 |
| 2. My unit has enough workers with the right training and skills to do their job in the best possible way. | 52 | 2 | 46 |
| 3. My unit has enough space to provide healthcare services. | 51 | 2 | 47 |
| 4. My unit has access to the transport and fuel that are needed to provide healthcare services. | 88 | 0 | 13 |
| 5. My unit has access to the communication tools (eg, telephone or radio) that are needed to provide healthcare services. | 84 | 2 | 14 |
| 6. My unit has enough medicine to provide healthcare services. | 48 | 2 | 50 |
| 7. My unit has enough functional equipment, such as a thermometer and blood pressure cuff, to provide healthcare services. | 49 | 4 | 47 |
| 8. My unit has enough disposable medical equipment, such as syringes, gloves and needles, to provide healthcare services. | 30 | 0 | 70 |
| 9. If the workload increases, my unit can get additional resources such as medicine and equipment. | 45 | 2 | 53 |
| 10. My unit receives money according to an established financial plan. | 84 | 2 | 14 |
| 11. My unit has money that we can decide how to use. | 91 | 3 | 6 |
| Community engagement | | | |
| 12. In my unit we ask community members what they think about the healthcare services that we provide. | 7 | 0 | 93 |
| 13. In my unit we listen to what community members think about the healthcare services we provide. | 4 | 1 | 95 |
| 14. In my unit we have meetings with community members to discuss health matters. | 5 | 1 | 93 |
| 15. In my unit we encourage community members to contribute to improving the health of the community. | 3 | 1 | 96 |
| 16. In my unit we encourage other organisations to contribute to improving the health of the community. | 11 | 0 | 89 |
| Monitoring services for action | | | |
| 17. I receive regular updates about my unit's performance based on information/data collected from our unit. | 14 | 3 | 84 |
| 18. My unit discusses information/data from our unit in a regular, formal way, such as in regularly scheduled meetings. | 11 | 7 | 82 |
| 19. My unit regularly uses unit information/data to make plans for improving its healthcare services. | 13 | 4 | 84 |
| 20. My unit regularly monitors its work by comparing it with the unit's action plans. | 13 | 5 | 83 |
| 21. My unit regularly compares its work with national or other guidelines. | 16 | 2 | 82 |
| Commitment to work | | | |
| 27. I am proud to work in this unit. | 21 | 3 | 76 |
| 28. I am satisfied to work in this unit. | 16 | 5 | 80 |
| 29. I feel encouraged to do my very best at work. | 7 | 3 | 89 |
| Work culture | | | |
| 30. My unit is willing to use new healthcare practices such as guidelines and recommendations. | 4 | 1 | 95 |
| 31. My unit helps me to improve and develop my skills. | 28 | 2 | 70 |
| 32. I am encouraged to seek new information on healthcare practices. | 20 | 3 | 78 |
| 33. My unit works for the good of the clients and puts their needs first. | 6 | 1 | 93 |
| 34. Members of the unit feel personally responsible for improving healthcare services. | 6 | 0 | 94 |



**Table 4**  Continued

| Resource | Disagree | Neutral | Agree |
|---|---|---|---|
| 35. Members of the unit approach clients with respect. | 2 | 2 | 96 |
| Leadership | | | |
| 36. I trust the unit leader. | 7 | 3 | 91 |
| 37. The leader handles stressful situations calmly. | 12 | 4 | 84 |
| 38. The leader actively listens, acknowledges, and then responds to requests and concerns. | 11 | 4 | 85 |
| 39. The leader effectively resolves any conflicts that arise. | 14 | 5 | 82 |
| 40. The leader encourages the introduction of new ideas and practices. | 13 | 4 | 83 |
| 41. The leader makes things happen. | 11 | 5 | 85 |
| Informal payment | | | |
| 42. Clients must always give informal payment to health workers to access healthcare services. | 97 | 1 | 3 |
| 43. Clients are treated more quickly if they make informal payments to health workers. | 98 | 0 | 2 |
| 44. Medicines or equipment that should be available for free to clients have been sold in my unit. | 97 | 1 | 3 |
| 45. Health workers are sometimes absent from work earning money at other places. | 97 | 1 | 2 |
| 46. Health workers in my unit give healthcare services to friends and family first. | 95 | 1 | 3 |
| 47. Health workers in my unit give jobs or other benefits to friends and family first. | 97 | 1 | 3 |
| 48. Efforts are made to stop clients from providing informal payment to get appropriate healthcare services. | 21 | 7 | 72 |
| 49. Efforts are made to stop health workers from asking clients for informal payment. | 21 | 8 | 71 |

information sources and a severe lack of resources to perform their work. The latter was also reflected in the health post preparedness assessment, which overall was on a low level.

So far, there is no consensus on defining or assessing the health system context.[33] Several contextual factors are associated with quality improvement, like leadership, organisational culture, information system and organisational structure. However, there are uncertainties regarding definitions and measurements.[34] Qualitative studies have contributed to the understanding of the health system context and quality of care. In this study, we quantified the perceived context and compared it with

health post preparedness. The COACH tool was developed in five countries. Later, it has been used in Mozambique,[35] and now in four different Ethiopian regions and three languages. Except for the sources of knowledge dimension, all other COACH tool dimensions showed good internal reliability. Items included in *source of knowledge* might not measure the same construct. The original validation process of this tool also indicated a low internal reliability of this dimension.[11] Some of the included items may not be relevant in certain contexts. We suggest this dimension with its five items for further evaluation.

The health extension workers lacked sources of new knowledge. Internet, e-health or m-health applications

**Table 5**  Percentage of items and dimensions in source of knowledge of the Context Assessment for Community Health tool in four Ethiopian regions, 2018 (N=152)

| Sources of knowledge | Not available | Never, 0 times | Rarely, 1–5 times | Occasionally, 6–10 times | Frequently, 11–15 times | Almost always, 16 times or more |
|---|---|---|---|---|---|---|
| 22. Clinical practice guidelines | 15 | 4 | 18 | 16 | 16 | 32 |
| 23. Other printed material for work (eg, textbooks, journals) | 18 | 9 | 25 | 30 | 14 | 5 |
| 24. The Internet | 68 | 25 | 5 | 1 | 0 | 1 |
| 25. Electronic decision support (eg, mobile phone applications or other electronic devices to assist with care and decision-making) | 56 | 30 | 3 | 5 | 5 | 2 |
| 26. In-service training/workshops/courses | 25 | 14 | 19 | 28 | 10 | 4 |

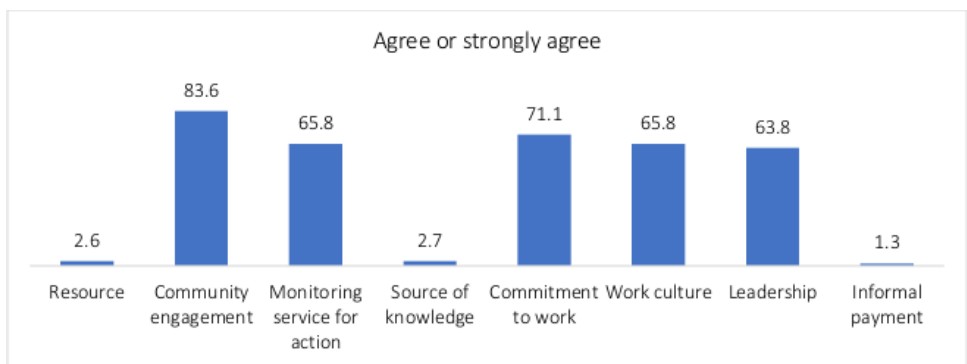

**Figure 1** Per cent agreement to dimensions in the Context Assessment for Community Health tool. Health extension workers in four Ethiopian regions, 2018 (N=152).

were absent.[35] Importantly, their responses indicated that they lacked in-service training, workshops and courses. Insufficient sources of knowledge could lead to inappropriate diagnosis and mismanagement, such as the irrational use of antibiotics. An earlier study conducted in the same study area indicated that the health extension workers' clinical assessment, classification and management of sick children did not follow the clinical guidelines.[22] This low adherence could lead to misdiagnoses and a lack of potentially life-saving treatments. Capacity building could be achieved through refresher training, followed by supportive supervision.

The health extension workers reportedly had good contact with the community they served. This engagement could help to enhance the health extension workers' accountability and dedication. A study conducted in southern Ethiopia indicated that with focused training, guidance and regular supportive supervision, the health extension workers enhanced in community participation.[36] A qualitative study in southern Ethiopia revealed that health extension workers' relationships with the community could be constrained due to inadequate support systems, trust, communication and dialogue, as well as differing expectations.[37] A study conducted in six regions of Ethiopia indicated that there were challenges in work schedule and relationship with the community.[38]

We also found that commitment to work was relatively good. A combination of financial and non-financial incentives

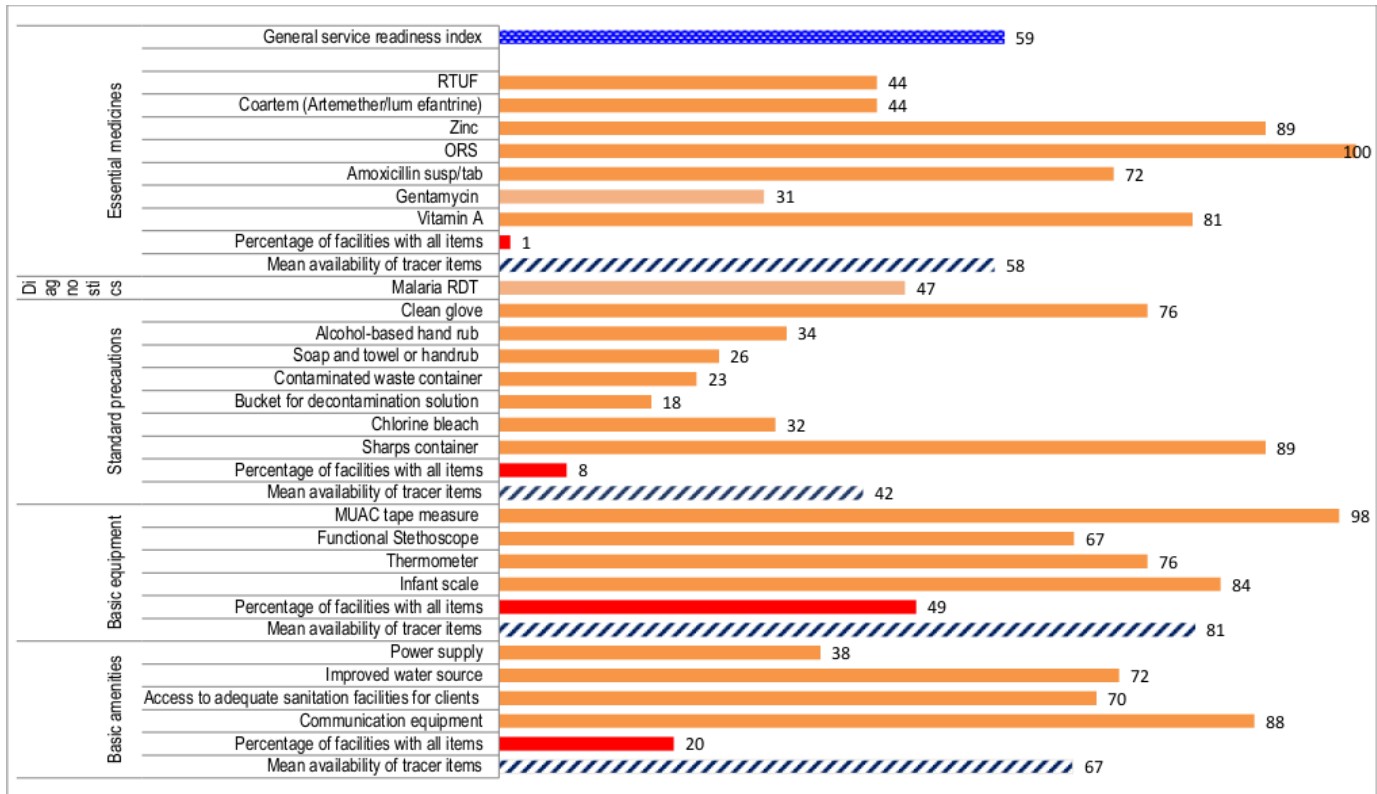

**Figure 2** Percentage of health posts with tracer items available to provide general service in four regions of Ethiopia, 2018 (N=152). MUAC, mid-upper arm circumference; ORS, oral rehydration solution; RDT, rapid diagnostic test; RTUF, ready-to-use therapeutic food.

is required to support motivation and satisfaction.[39] Non-financial incentives, such as creating career opportunities, may increase the motivation and retention of health extension workers.[40] Over the course of a week, the health extension workers spend their time at the health post (51%), in the community (37%) and elsewhere, unable to get information (12%).[15] This is an indication that they spend most of their time with the community that has a potential to enhance the interpersonal communication.

Informal payments were perceived to be very rare. Informal payments or various forms of corruption could have significant adverse effects on the health system, affecting patients and service providers.[41] A study in Tanzania showed that informal payments existed and were negatively associated with job satisfaction and motivation.[42]

The health posts in the study area seemed to have moderate service readiness, especially basic amenities and equipment. However, health posts did not have the essential medicines to provide child care. These facilities are the first contact for primary care, which provides basic health and medical care close to the community, especially in rural populations.[19] These first-line services can potentially respond to a range of health challenges in low-income countries,[43] but only if proper attention is given to needs, such as essential medicines, in addition to infrastructure and basic equipment. To meet such requirements is vital for a resilient health system.[44]

We have earlier shown that health extension workers' ability to classify childhood illnesses was low.[23] The evaluation of the Optimizing the Health Extension Program intervention's effectiveness showed no effect on the utilisation of services for sick children.[24] The lack of effect could partly be attributed to delays, interruptions and an overall short implementation period of a complex intervention. Complex interventions that aim to change health services and care-seeking for sick children may need an extended implementation period.[45] Lack of effect could also be due to some of these contextual factors necessary for improving quality.

The context in which the services are provided is essential for implementing changes or new programmes. However, contextual factors are generally not well understood. Before this study, no assessments of different aspects of the health system context have been done in Ethiopia. A study conducted in six European countries found that structure and process indicators explained more variability in client satisfaction than contextual factors.[46] A systematic review concluded that contextual factors might influence the effectiveness of quality improvement interventions.[47]

We report the first study in Ethiopia of primary healthcare workers' perceived health system context. The COACH tool has been validated in a range of other low-income countries[11] and was also found to have satisfactory internal reliability when translated into three Ethiopian languages. Understanding context can identify factors that promote or hinder the implementation of evidence-based practices, increasing the likelihood of successful implementation. Although precautions were taken to obtain valid responses from the interviewed health extension workers, the results could be susceptible to bias. The sample represented a large number of districts in four Ethiopian regions that participated in a child health services study, but inferences cannot be drawn to the whole country.

## CONCLUSION

The Ethiopian health extension workers' perceived context showed a severe lack of resources. They perceived a good relationship with the local community, used data for planning but lacked access to new knowledge. They were highly committed to work and had positive perceptions of their work culture and a relatively positive attitude regarding their leaders. There was no corruption or informal payments at their work sites. The internal consistency of the context assessment tool provided evidence of its ability to measure its different dimensions. This feature will allow for tailoring implementation strategies and assessing context as part of evaluations. The health extension workers' perceptions of sources of information and available resources were in line with the results of the health facility preparedness.

**Acknowledgements** The authors would like to thank the field teams that were involved in the data collection as well as the government officials who facilitated the administration of the surveys. The authors would also like to thank the study participants who agreed to give their time to participate in the study.

**Contributors** TG, SMA, MY, L-AP and DB conceptualised the design of the study. TG analysed the data. TG, SMA, MY, AB, L-AP and DB provided review of the methodology and interpreted the results. All authors contributed to the writing of this paper and all have read and approved the final manuscript.

**Funding** This project was funded by Bill & Melinda Gates Foundation (INV-009691).

**Competing interests** None declared.

**Patient consent for publication** Not required.

**Ethics approval** Ethical approvals were obtained from the University of Gondar (Ref O/V/P/RCS/05/371/2018), the Ethiopian Public Health Institute (Ref 613/52), and the London School of Hygiene and Tropical Medicine (Ref 16117). Information sheets were translated into the local languages Amharic, Oromiffaa and Tigrigna, and read to obtain written informed consent.

**Provenance and peer review** Not commissioned; externally peer reviewed.

**Data availability statement** Data are available upon reasonable request. Request for data can be made to DB (della.berhanu@lshtm.ac.uk). Data sharing policy has been developed. All requests will be reviewed by data sharing committee and if granted, data will be shared without any identifiers.

**ORCID iDs**
Theodros Getachew http://orcid.org/0000-0002-6486-0531
Solomon Mekonnen Abebe http://orcid.org/0000-0002-1150-2319
Lars-Ake Persson http://orcid.org/0000-0003-0710-7954
Della Berhanu http://orcid.org/0000-0002-4984-893X

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
