## [Reviewer comments · BMJ Open]

ARTICLE DETAILS

TITLE (PROVISIONAL)	Health extension workers' perceived health system context and health post preparedness to provide services: A cross-sectional study in four Ethiopian regions
AUTHORS	Getachew, Theodros; Abebe, Solomon; Yitayal, Mezgebu; Bergström, Anna; Persson, Lars; Berhanu, Della

VERSION 1 – REVIEW

REVIEWER	Fowler Davis, Sally Sheffield Hallam University, Faculty of health and Wellbeing
REVIEW RETURNED	03-Mar-2021

GENERAL COMMENTS	Thank you for the opportunity to review the paper which is novel and important in respect of understanding the context of implementation. I was impressed by the rigor of the processes used to translate and use existing tools in a different culture and language. I have few recommendations as the paper is presented and argued well, however the sampling is somewhat unclear and needs to be clarified. 200 enumeration areas selected with one worker interviewed and context analysed but only 165 'health posts' assessed - are these the contexts?. Similarly 154 workers interviewed so please include more information about what happened to those areas and persons not included- where they the same- suggesting that some areas were excluded from the data altogether or where there different and often additional areas of non-participation? Without further explanation it calls into question the statistical method that seeks to quantify the determinants of context readiness. In addition , the discussion should include some further details about the Ethiopian health system to enable the reader to qualify assertions associated with 'good contact with the community' and 'commitment to work was relatively good' . I would expect some more detailed narrative about the relative impact on the overall quality of services and the presence or otherwise of systems and processes to performance manage these services. To strengthen the discussion there is a need to add some detail about the processes associated with worker experience-management/ contractual arrangements and such. It is too limited to suggest that trust and communication are affective when you clearly have reference to other elements of the study to provide further narrative and nuance within this paper.
---

	Use of STroBE is helpful but 16 b and c would be useful additions
--	---

REVIEWER	Martin, Steven University of East London - Stratford Campus
REVIEW RETURNED	06-Mar-2021

GENERAL COMMENTS	Congratulations to the authors on their paper, I feel this original research contributes to a better understanding of the primary health care system in Ethiopia and to the development of an insightful tool; however, there are some revisions required. Abstract/Introduction/general  - Authors need to go through the BMJ Open formatting guide and update referencing accordingly. For example, "quality-of-care,[1]." not "quality-of-care (1)." (line 6, page 5) - I understand that BMJ Open uses British English and therefore American English should be avoided (for example, line 18, page 5 – “operationalisation” not “operationalization”) - Please amend the text “under-five children’s primary level services” to improve readability (line 40 page 5). Some suggested text can be found in as a comment in the attached PDF. - Please provide a clear definition of what constitutes a ‘health post’ (line 45/46 page 5) - Please amend the text “under-five health services” to improve readability (line 12/13 page 6). Some suggested text can be found in as a comment in the attached PDF. - Please add the Getachew T et al. reference to the bibliography (line 26 page 6) Methods  - Please remove the words “This tool, labelled” and start the sentence with “The” (line 24 page 7) - Could further details be provided on the linguistic issues? For example, which words or concepts were difficult to translate? Were there issues with translating ‘sources of knowledge’ or ‘community engagement (lines 46-47: page 7)? - Could further details be provided on the translation process? For example, was this done by the research team or was external support provided? (for example, lines 38-42 page 7, and lines 33-36 page 8) Findings  - Table 4 should be reformatted if the findings from ‘sources of knowledge’ are being presented differently. Perhaps a separate table? Discussion  - Could the authors discuss the implications for the tool if the internal consistency of source of knowledge is insufficient? (lines 38-40, page 14)
---

VERSION 1 – AUTHOR RESPONSE

Reviewer: 1

Dr. Sally Fowler Davis, Sheffield Hallam University, Sheffield Teaching Hospitals NHS Foundation Trust

Comments to the Author:

Thank you for the opportunity to review the paper which is novel and important in respect of understanding the context of implementation. I was impressed by the rigor of the processes used to translate and use existing tools in a different culture and language.

1. I have few recommendations as the paper is presented and argued well, however the sampling is somewhat unclear and needs to be clarified. 200 enumeration areas selected with one worker interviewed and context analysed but only 165 'health posts' assessed - are these the contexts?. Similarly 154 workers interviewed so please include more information about what happened to those areas and persons not included- where they the same- suggesting that some areas were excluded from the data altogether or where there different and often additional areas of non-participation?

Response: Thank you. We have added a description on the result section. Page 9 line 178 as follows. Of the 200 enumeration areas, 20 were not included due to local unrest. The remaining 180 enumeration areas were served by 165 health posts. A total of 165 health posts were assessed, and 154 health extension workers were available for interview.

2. Without further explanation it calls into question the statistical method that seeks to quantify the determinants of context readiness.

Response: Thank you. Maybe you refer to the statement "Mapping the facility preparedness sets the scene, but such assessments are poorly associated with the quality of services provided"? This statement is backed up by findings from a multi-country study with such a result. Page 4 line 88.

3. In addition , the discussion should include some further details about the Ethiopian health system to enable the reader to qualify assertions associated with 'good contact with the community' and 'commitment to work was relatively good' . I would expect some more detailed narrative about the relative impact on the overall quality of services and the presence or otherwise of systems and processes to performance manage these services.

Response: Thank you. These statements are based on the findings from our study, reflecting the perceptions of the health extension workers. As we state in the summary of findings in the introduction of the Discussion: "The health extension workers perceived that they had a good relationship with the community", and "...were highly committed to their work". We have also added a brief explanation to the existing policy on the health extension programme and health extension workers. Page 4 line 96.

4. To strengthen the discussion there is a need to add some detail about the processes associated with worker experience- management/ contractual arrangements and such. It is too limited to suggest that trust and communication are affective when you clearly have reference to other elements of the study to provide further narrative and nuance within this paper.

Response: Thank you. Added. Page 4 line 96 and 98. The health extension workers training has been explained as follows.

The health extension workers are recruited from the community they serve and deployed to service after a 1-year formal pre-service training provided after completing 10th grade of formal education

Use of STROBE is helpful but 16 b and c would be useful additions

Response: Thank you. We don't find 16 b and c relevant with a study of this design (this is a cross-sectional study, no relative risk estimates, etc.) and we put 'Not Applicable' in STROBE checklist.

Reviewer: 2

Mr. Steven Martin, University of East London - Stratford Campus

Comments to the Author:

Congratulations to the authors on their paper, I feel this original research contributes to a better understanding of the primary health care system in Ethiopia and to the development of an insightful tool; however, there are some revisions required.

Abstract/Introduction/general

5. - Authors need to go through the BMJ Open formatting guide and update referencing accordingly. For example, "quality-of-care,[1]." not "quality-of-care (1)." (line 6, page 5)

Response: Thank you. We updated the referencing accordingly.

6. - I understand that BMJ Open uses British English and therefore American English should be avoided (for example, line 18, page 5 – “operationalisation” not “operationalization”)

Response: Thank you. Revised.

7. - Please amend the text “under-five children’s primary level services” to improve readability (line 40 page 5). Some suggested text can be found in as a comment in the attached PDF.

Response: Thank you. Changed as suggested.

8. - Please provide a clear definition of what constitutes a ‘health post’ (line 45/46 page 5)

Response: Thank you. We have added a sentence describing health post. Page 5 line 104.

9. - Please amend the text “under-five health services” to improve readability (line 12/13 page 6). Some suggested text can be found in as a comment in the attached PDF.

Response. Thanks. We have changed accordingly.

10. - Please add the Getachew T et al. reference to the bibliography (line 26 page 6)

Response: This paper has now been published and we have inserted an appropriate citation in the text as well as in the bibliography.

Methods

11. - Please remove the words “This tool, labelled” and start the sentence with “The” (line 24 page 7)

Response: Edited accordingly.

12. - Could further details be provided on the linguistic issues? For example, which words or concepts were difficult to translate? Were there issues with translating ‘sources of knowledge’ or ‘community engagement (lines 46-47: page 7)?

Response: Errors were not at sentence-level, but at word-level. We don’t find it meaningful to list these single-word errors and prefer to keep the general statement.

13. - Could further details be provided on the translation process? For example, was this done by the research team or was external support provided? (for example, lines 38-42 page 7, and lines 33-36 page 8)

Response: Thank you. The following sentence has been added: “The forward translation was done by a professional translator. The review, backward translation, and comparisons were done by a group of experts, including the study team.”. Page 7 line 149.

Findings

14. - Table 4 should be reformatted if the findings from ‘sources of knowledge’ are being presented differently. Perhaps a separate table?

Response: Thank you. This is now presented separately, Table 4 and 5.

Discussion

15. - Could the authors discuss the implications for the tool if the internal consistency of source of knowledge is insufficient? (lines 38-40, page 14)

Response: Thank you. We have added this to the discussion. Page 14 line 253.

VERSION 2 – REVIEW

REVIEWER	Martin, Steven University of East London - Stratford Campus
REVIEW RETURNED	05-May-2021
GENERAL COMMENTS	Congratulations to the authors on their paper, I feel this original research contributes to a better understanding of the primary health care system in Ethiopia and to the development of an insightful tool. I would also like to thank the authors for addressing some of my comments in the latest version.